Fairness-based techniques to optimize vaccine allocation among migrants during pandemics: a scoping review

Şimşek Sera serasimsek42@gmail.com 1
Altay Sevval 2
Salman F. Sibel 3
Kayı İlker 1
1 Department of Public Health, Koc University , Istanbul , Turkey
2 Koc University , Istanbul , Turkey
3 Industrial Engineering Department, Koc University , Istanbul , Turkey
Geard Nicholas
Electronic publication date: 2025 Oct 30
Publication date: 2025
Volume: 13
Electronic Location ID: e20208
Received 2025 Apr 23; Accepted 2025 Sep 18
Copyright: ©2025 Şimşek et al.
Copyright year: 2025
Copyright holder: Şimşek et al.
License: This is an open access article distributed under the terms of the Creative Commons Attribution License, which permits unrestricted use, distribution, reproduction and adaptation in any medium and for any purpose provided that it is properly attributed. For attribution, the original author(s), title, publication source (PeerJ) and either DOI or URL of the article must be cited.
License URL: https://creativecommons.org/licenses/by/4.0/

Keywords: Vaccine, Fairness, Pandemi, Migrant

Funding: The authors received no funding for this work.

==============================
Introduction

Migrants face significant barriers to vaccination due to disparities in access and coverage, necessitating fairness-based strategies and inclusive healthcare infrastructure to ensure equitable immunization, especially during pandemics. This study investigates fairness-based vaccination strategies, focusing on migrant vaccination status during pandemics, and migrant specific vaccine distribution models.

Methods

The authors employed established scoping review methods to explore the research question: How have fairness-based strategies for vaccine allocation affected vaccination coverage among migrants during pandemics in urban and rural areas? A scoping review was conducted following the PRISMA and expectation, client group, location, impact, professionals, and service (ECLIPSE) guidelines, utilizing the Joanna Briggs Institute’s Checklist for Qualitative Research. The review involved a comprehensive database search across PubMed, Scopus, Web of Science, Cochrane Library, and Ovid MedLine. The eligibility criteria for publications included at least one of the following aspects related to migrants: access to vaccines or frequency of vaccine uptake, vaccine hesitancy, vaccine modeling and optimization approaches, or discussions grounded in principles of fairness. Searches were limited to the articles published in English between 2000-2022. Initially, 5,653 articles were identified, which were reduced to 305 after title screening. Following abstract screening, 19 articles meeting the inclusion criteria—focused on vaccination modeling, allocation, fairness optimization, and behaviors or attitudes in migrant populations—were selected for full-text evaluation.

Results

Vaccination rates among migrants range from 42.7% to 87%, which are lower compared to the host population. Although the willingness to vaccinate is around 70%, significant barriers such as language obstacles, lack of access to healthcare services, and insufficient information remain critical challenges. While 19 of the studies defined fairness through the use of health services, four of them discussed it on community participation, and two employed modeling approaches. Various techniques, including community involvement, digital health messages and national refugee centers, have been employed to allocate vaccines fairly and consistently. The concept of equity has been addressed inconsistently across studies, and there is insufficient data to develop a fair vaccine distribution strategy for migrant populations.

Conclusion

This study highlights the following: (1) the challenges migrants face, including limited access to healthcare, language barriers and poor living conditions, which complicate equitable vaccine allocation; (2) the lack of specific, systematic national vaccine allocation programs targeting migrants; and (3) the need for a targeted, fairness-based approach, along with further research on national policies and vaccine delivery models that prioritize migrants and address their unique vulnerabilities.

Introduction

The term refugee is defined by international law as individuals who, owing to a well-founded fear of persecution based on race, religion, nationality, membership in a particular social group, or political opinion, are outside their country of nationality and unable or unwilling to return or seek that country’s protection (UN Refugee Agency, 1951). In contrast, migrants refer to people who move from one place to another, whether across or within international boundaries often for economic, educational or family reasons. Asylum seekers are individuals who seek international protection but whose refugee status has not yet been legally determined. Collectively, these groups fall under the broader category of forcibly displaced persons, which includes refugees, asylum seekers, and internally displaced people (UN Refugee Agency, 2021). According to the UN Refugee Agency, 27.1 million people worldwide have been globally displaced in 2021 (World Health Organization, 2022). The difficulties these refugees face, such as poverty, lack of access to preventive healthcare—especially vaccination—poor living conditions and language barriers, necessitate developing adequate strategies related to healthcare services to alleviate such dire conditions (Jasko et al., 2021). Disparities in the vaccine-preventable disease burden and immunization coverage between refugees and the host population have been identified in many countries worldwide (Charania et al., 2020). Refugees generally have a higher vaccine-preventable disease burden and lower immunization rates than the host population (Charania et al., 2019). During the COVID-19 pandemic the entire refugee group, especially refugee workers who are the main workforce in the countries they have migrated to, had a high risk of infection (Brickhill-Atkinson & Hauck, 2021). In addition, access to healthcare was inevitably difficult for those living in rural areas or urban “hot spot” areas. In particular, the presence of a mobile refugee population in transitional countries such as Turkey increases the trend towards mobile service models and complicates the access problem (Bayraktar et al., 2022).

Variations in immunization rates among refugees—often associated with factors such as place of birth, cultural belief, language proficiency, migration history, and length of residency—highlight the systemic and contextual barriers that hinder equitable service delivery (Salman et al., 2021). For vaccination of refugees, priority areas must be created in the current health system, or alternative and effective strategies should be explored (Charania et al., 2020). Fairness-based vaccine allocation refers to strategies that prioritize equitable access to vaccines, ensuring that disadvantaged or high-risk groups receive appropriate consideration during distribution, especially in resource-limited pandemic settings.

Given the critical role of vaccination during pandemics, achieving inclusive immunization requires a technical infrastructure that minimizes access barriers for migrants. Fairness-based vaccination is essential to reduce inequities and ensure equitable vaccine access, especially for marginalized groups. However, fairness of vaccine allocation has not been adequately explored in the literature. Researchers have identified four key approaches to fairness: ensuring equal treatment for all, prioritizing those who are most vulnerable, maximizing overall benefits, and encouraging or rewarding social utility (Yi & Marathe, 2015; Rumpler et al., 2023; Do et al., 2023). Research in this area has often relied on subjective criteria, such as efficiency and ethics, without delving into operational strategies (Yi & Marathe, 2015).

Fairness-based vaccination is defined as vaccine strategies specifically designed for vulnerable or marginalized groups, aimed at maximizing effectiveness, efficiency, and access to achieve community immunization (World Health Organization, 2020). While fairness and equity are often used interchangeably, this study distinguishes them to clarify their roles in vaccination strategy. Equity emphasizes distribution based on needs and vulnerabilities, aiming to reduce disparities and ensure equal health outcomes. In contrast, fairness refers to the legitimacy, transparency, and consistency of the decision-making process. It involves ensuring that vaccination policies are applied consistently, free from bias or favoritism, and are perceived as just by the public. While equitable policies seek to achieve equal health outcomes, fair processes strive to ensure that the criteria and procedures behind vaccine allocation are morally and socially acceptable. On the other hand, in the case of migrants, who are often mobile, undocumented, or outside of standard systems, focusing solely on outcomes may be insufficient (World Health Organization, 2020; Prentice et al., 2024; Persad, Peek & Ezekiel, 2020; Dai et al., 2025).

Vaccination campaigns in response to pandemics target providing the most protection to the greatest number of people in the least amount of time by maintaining an efficient and effective system for allocating, distributing, and administering vaccines (Muckstadt et al., 2023). The vaccine allocation problem involves determining the most equitable and efficient distribution of limited vaccine supplies among various populations (Erdoğan et al., 2024). The impact of these decisions is evaluated by different mathematical models. In simulating the spread of disease in a population and evaluating the effectiveness of different vaccine allocation strategies, modified versions of the classical susceptible, infective, recovered (SIR) model have been used (Erdoğan et al., 2024). Several extensions of the SIR model have been proposed for the COVID-19 disease. One commonly used extension of the SIR model is the SEIR model, which includes four compartments: S (susceptible), E (exposed), I (infectious), and R (recovered), which has been extended further to represent the transition dynamics of COVID-19, taking into account the effects of vaccination as well.  System dynamics and agent-based simulation are alternative approaches to evaluate the effectiveness of vaccination strategies. The main principles in allocating COVID-19 vaccines have been preventing harm, prioritizing people who are disadvantaged, and achieving equal treatment (Persad, Peek & Emanuel, 2020). In the distribution and administration of vaccines to the public, several types of facilities have been utilized: large vaccination centers in urban areas, hospitals and local clinics and mobile facilities for rural and remote areas.

Despite increasing efforts to include refugees and migrants in public health policies, many have still faced significant barriers to vaccination and accessing health systems necessary for the administration of COVID-19 vaccines (Immordino et al., 2022). As stated by Immordino et al., these barriers encompass skepticism about the vaccine’s safety and benefits, societal norms and pressures, inadequate information on obtaining vaccines, language difficulties, complicated registration procedures, restricted internet access and concerns about arrest, detention, or deportation. Extra efforts are needed to overcome these barriers.

The aim of this study is to determine fairness-based vaccination strategies, the vaccination frequency of migrants, the vaccination policies (programs) on migrants during a pandemic, and vaccine distribution models specific to migrant populations. In the long term, it will contribute to the development of inclusive and fair vaccine distribution models capable of addressing the unique challenges faced by migrants and providing valuable guidance for public health policies. This scoping review is intended for public health professionals, policymakers, humanitarian organizations, and researchers seeking evidence-based insights into fairness-oriented vaccination approaches for migrant populations during pandemics.

Methods

Study populations

An immigrant is someone who chooses to leave their country voluntarily, often in search of better living conditions, economic opportunities, education or family reunification. A refugee is someone who is compelled to leave their country to escape war, persecution or natural disasters, and seeks protection in another country under international law. An asylum seeker is an individual who has applied for refugee status and is awaiting a legal decision on whether they will be recognized as a refugee (UN Refugee Agency, 1951; UN Refugee Agency, 2021; World Health Organization, 2022). In this study, the groups included were collectively represented under the general term “migrants”, encompassing refugees, asylum seekers, and other forcibly displaced persons, reflecting the populations described in the included studies.

Assessment tools

In this study, we systematically investigated articles on the vaccination status of migrants, analyzing the percentage of vaccinated individuals and classifying their characteristics. We also reviewed research methods, outcomes, and findings from relevant studies to propose potential vaccine allocation strategies. To achieve this, we conducted a comprehensive literature review using the expectation, client group, location, impact, professionals, and service (ECLIPSE) framework, along with the PRISMA guidelines and the Joanna Briggs Institute’s Checklist for Qualitative Research (Page et al., 2021; Wildridge & Bell, 2002). The keywords obtained in studies were listed according to the ECLIPSE classification. The location step was eliminated because it was not relevant to our inquiry.

Literature search

While creating the search strategy of this study, the keywords “refugees” and “COVID-19 vaccine” were taken as a basis. At the same time, this base was expanded by including the words migrant, immigrant and asylum seeker.

The following terms were used: (vaccine OR vaccination OR Mass Vaccination OR Immunization OR Mass Immunization OR COVID-19 Vaccines OR COVID-19 Vaccination OR Influenza Vaccines OR Influenza Vaccination) AND (Public Health Emergency OR Public Health Service) AND (Prioritization OR Priority OR Rationing OR Routing OR Health Care Rationing OR Allocation OR Resource Allocation OR Fair allocation OR Equity OR Health Equity OR optimization) AND (Priority populations OR Risk groups OR at risk populations OR Migrants OR Refugees). The publications were obtained from searches made from PubMed, Scopus, Web of Science, Cochrane Library, and Ovid Medliner databases, restricted to articles published in English continuing all the key word between 2000–2022.

After this step, the articles were screened in two steps by two researchers. At each step, the articles selected by the researchers were combined, and duplications were eliminated. A critical evaluation of the quality of the selected articles was conducted by two researchers. This evaluation utilized the parameters outlined in the Joanna Briggs Institute’s Checklist for Qualitative Research (Lockwood, Munn & Porritt, 2015). These parameters provided a robust framework for assessing the reliability and validity of the research findings. The synthesis of the data was performed independently by two researchers, and any discrepancies were resolved through a joint review, with one researcher making the final decision.

The data was systematically extracted and categorized under the following headings: author(s), Year, Title, Country, Sample, Study Design, Objectives, Vaccine(s), Operation (Fairness-Based Techniques), and Result. This structured approach ensured clarity and consistency in organizing the information from the included studies.

Eligibility criteria

While mapping the scope of the research, the inclusion and exclusion criteria were determined by holding bi-weekly meetings with a three-member research group. Publications included in this study addressed at least one of the following criteria within communities of refugees or migrants:

• Access to vaccines or vaccine uptake frequency

• Vaccine hesitancy, including decision-making processes, attitudes or behaviors regarding vaccination

• Vaccine modeling studies targeting these groups

• Discussing vaccine allocation and, if available, optimization approaches

• Focusing on discussions grounded in fairness principles

Publications that did not focus on refugee or migrant groups or did not involve vaccination were excluded from the evaluation.

Data analysis

In the first step, 6,397 publications were identified. After removing duplicates with EndNote and the Systematic Review Accelerator (SRA) De-duplicate tool, and excluding records marked as ineligible by automation tools such as Abstrackr and Rayyan, 5,653 publications remained. In the second step, the title screening yielded 305 articles, which were independently reviewed by two researchers (SS and SA) based on their abstracts. Discrepancies were resolved through consensus. After abstract screening, 60 articles were identified, of which 47 met the inclusion criteria following independent review by two researchers (SS and SA) and were selected for full-text evaluation. Based on the full-text review, ten articles were excluded as they did not address issues of vaccine allocation, while three were excluded as they did not discuss issues of migrants. Among the 47 studies that met the inclusion criteria, 27 were further excluded because they were reviews, viewpoints, reports, quick reviews, or thorough reviews, and one qualitative study was excluded because it collected opinions on migrants’ vaccinations exclusively from healthcare workers in the vaccination field, without directly including migrants. At the end of the literature review, 19 articles were finalized to be included in the study. The review group followed the PRISMA reporting guideline for the reporting of this scoping review (Fig. 1).

Figure 1 PRISMA flow diagram of the review process.

Results

Characteristics of included studies

Data on the following topics were taken from each study: study method: quantitative, qualitative, mixed methods; study design: cross-sectional, intervention, modeling; study sample: refugee, immigrant, migrant, and mixed with host population; operation (fairness-based techniques); and results (significance, effect size). Nineteen operational studies out of the 47 total included in the study are displayed in the findings section. The remaining 27 articles are reviews, viewpoints, reports, quick reviews, thorough reviews and scoping reviews, with one being qualitative without directly including migrants. Nineteen articles were found focusing specifically on operational studies examining the fairness-based allocation of vaccines in migrants.

The words migrant and immigrant were observed the most in the population; modeling studies were followed most frequently, and as an outcome, accept(ance) was mentioned more frequently than fairness and equity. COVID-19 was mentioned in two-thirds of the studies (Table 1). Out of the 5,653 appearances in the records, only 14 publications included the term ”fairness”, accounting for just 0.2% of the entire dataset. The term “refugee” appeared in 1.6% of publications, while ”(im)migrant” was found in 2.5%. The terms “allocation” and ”optimal” or “optimization” appeared at similar rates, whereas ”model(ing)” and ”access to health” were found in 3.8% and 12.4% of titles, respectively. The terms “acceptance” and “hesitancy” occurred at comparable rates.

Table 1 Key framework terms identified in the literature.

Code/thematic area	Term	Number of appearances in records (n = 5,653)	
Expectation
(improvement or information or innovation)	Fair(ness)	14 (0.2)	
Equity	47 (0.8)	
COVID-19	1,750 (30.9)	
Influenza	1,035 (18.3)	
Vaccine(ation)	1,831 (32.3)	
Pandemic	825 (14.5)	
Immunization	128 (2.2)	
Client group
( at whom the service is aimed)	Refugee	96 (1.6)	
(Im)Migrant	146 (2.5)	
Minority	29 (0.5)	
Vulnerable	36 (0.6)	
Impact
( outcomes)	Allocation	56 (0.9)	
Optimal(ization)	46 (0.8)	
Management	111 (1.9)	
Model(ling)	216 (3.8)	
Access to health	701 (12.4)	
Professionals, and Service
( for which service are you looking for information)	Behaviour	28 (0.4)	
Accept(ance)	72 (1.2)	
Hesitancy	67 (1.1)	
Attitude	83 (1.4)	

Vaccination outcomes

Table 2 presents the characteristics and findings of both operational studies and review studies that met the inclusion criteria. The vaccinations “COVID-19” and “Other” come in two different varieties. In contrast, only the COVID-19 case was the subject of 35 investigations; 12 studies covered influenza, hepatitis B, measles, tetanus, etc. Three categories were recognized as migrants throughout all the studies: refugees, immigrants and migrants, and asylum seekers. Twelve studies focused on vaccination hesitancy, six examined vaccine access, and 13 examined vaccine uptake. Even though 27 of the studies were reviews, 14 had policy as their conclusion. On the distribution of vaccines among migrants, there are just two modeling studies. Vaccination rate was between 42.7% and 87% (Fig. 2). Most articles identified systematic, scoping, and rapid reviews and found that migrants experienced a higher burden of infection and lower vaccination rates than the host population. These studies also concluded that the desire to be vaccinated was around 70 percent, and reasons for not being vaccinated were listed as the lack of access to health services, health communication barriers (language), lack of support, feelings of being excluded in health interventions compared to the general population and lack of access to information. Under the title of Vaccination Strategy for Migrants, 8.5% of the articles emphasized the prioritization of migrant workers, while only 2.1% focused on the prioritization of elderly migrants. Vaccination campaigns in multiple languages were discussed in 6.3% of the publications. Specific campaigns targeting migrants were highlighted in 14.8% of the studies, and 10.6% addressed vaccination strategies for undocumented migrants.

Table 2 Characteristics of the included operational studies and the excluded review papers.

Characteristic of studies	Number of study (n = 47)	
Infection n (%)	COVID-19	35 (74.4)	
Other (Influenza, Hepatitis B, measles.)	12 (25.5)	
Stated population
n (%)	Refugees	19 (41.3)	
(Im) Migrant	34 (72.3)	
Asylum seeker	2 (4.3)	
Study design n (%)	Review	27 (57.4)	
Cross-sectional	13 (27.6)	
Qualitative	4 (8.5)	
Modeling study	2 (4.2)	
Intervention	1 (2.1)	
Outcome n (%)	Uptake	13 (27.6)	
Access to vaccine	6 (12.7)	
Vaccine hesitancy	12 (25.5)	
Policy	14 (29.7)	
Modelling	2 (4.2)	
Vaccination strategy for refugees n (%)	Prioritization of migrants workers	4 (8.5)	
Prioritization of elderly migrant	1 (2.1)	
Vaccination campaigns in multiple languages	3 (6.3)	
Specific vaccination campaigns targeting migrants	7 (14.8)	
Vaccination undocumented migrants	5 (10.6)	
Vaccination rate (%)		42.7–87.0	

Figure 2 Vaccination rates for refugee and migrant groups: an overview of relevant research.

Figure 2 illustrates vaccination rates reported in multiple studies focusing on refugee and migrant populations. Migrant vaccination rates ranged from 60% to 87%, whereas refugee vaccination rates ranged from 42.7% to 76.0%.

Fairness-based strategies

We defined how frequently factors such as acceptance, accessibility, and hesitancy have been investigated over time in the context of refugee vaccination during pandemics. Notably, acceptance shows a marked increase in recent years, reflecting growing research attention to vaccine uptake challenges. Overall, it underscores the evolving focus on fairness-based strategies in public health research concerning refugees.

Table 3 Characteristics of the included operational studies—strategies to defined vaccination of migrants.

Study design
Cross section	Study sample N	Operation (Fairness-based
Techniques)	Result	
Gilder et al. (2022)	253 postpartum migrant women	Contact by phone or home visit	86.8% vaccine completion	
Allen et al. (2022b)	300 migrants	Community leaders and
social media community
groups supporting migrants	87% vaccine completion
Actors associated with willingness were age (aOR 1.07) and no exposure to concerning news about COVID-19 vaccines (aOR 3.71).	
Zhang et al. (2021)	435 refugee	Message or e-mail an anonymous online survey	70% intention to vaccinate Being an essential worker (aOR: 2.3) and male sex (aOR: 1.8) are risk factors	
Elharake, Omer & Schwartz (2022)	20 LMICs and
20 HICs	N/A	13 LMICs specified standing nationwide routine immunization policies for refugees, while 14 HICs included refugees in their national routine immunization programs	
Führer et al. (2022)	204 migrants	N/A	80% vaccination rate Unvaccinated respondents feared side effects, were convinced that the vaccine was not safe, and assumed that COVID-19 was not dangerous	
Salibi et al. (2021)	3.838 Syrian refugees	An international
humanitarian
organization, Norwegian
Refugee Council	29% no intention to vaccinate Vaccine safety (aOR: 5.97) and effectiveness (aOR: 6.80)	
Alabdulla et al. (2021)	7,859 Qatari and migrant	N/A	20.2% would not vaccinate and 19.8% being unsure about taking COVID-19 vaccine
Citizens and females were more likely to be vaccine hesitators than immigrants and males	
Vita et al. (2019)	3,941 migrants	N/A	85% vaccine completion
The average of 10.5% of mi
grants vaccinated in the first three years to 66% in the last year	
Diaz, Dimka & Mamelund (2022)	1.284 migrants
4,158 non migrants	N/A	Fewer migrants than non-migrants reported receiving a vaccine offer (68.1% vs. 81.1%)	
Al-Hatamleh et al. (2022)	501 Palestinian
refugees, 491 Jordanian citizens	N/A	Compared to the citizens, the refugees had significantly lower levels of beliefs about the safety	
Teng, Hanibuchi & Nakaya (2022)	1,455 migrant	Psychological, linguistic,
economic, political,
social, navigational
integration	11.6% hesitancy Highly integrated migrants were re-
ported to have less vaccine hesitancy	
Allen et al. (2022a)	353 Brazilian immigrant women	The Health Belief Model	70.8% intention to vaccinate	
Debela, Garrett & Charania (2022)	178 refugee parents in New Zealand	N/A	21% of parents had delayed and 12% had refused to
vaccinate their child	
Intervention	
Streuli et al. (2021)	60 adult Somali
refugees and 7 expert advisors	Community-based
participatory research
(CBPR) models
Developing an innovative
vaccine educational
technology (VR)	CBPR approach can be effectively used for the codesign of a VR educational program
Cultural and linguistic sensitivities are essential factors for effective community engagement	
Qualitative	
Kowal, Jardine & Bubela (2015)	37 refugee	Community partner,
the Multicultural
Health Brokers of
Edmonton (MCHB)	No anti-vaccination sentiment
The lack of reach of
public health vaccination campaigns in Alberta.	
Mahimbo et al. (2022)	37 refugees	N/A	Cues for increasing individual willingness to get vaccinated included obtaining information from trusted sources and community engagement.	
Knights et al. (2021)	64 PCPs and
administrative staff, 17 recently- arrived migrants	Digital health messages	Digitalisation has language barriers, difficulties building trust, and the risk of missing safeguarding cues in virtual consultations	
Modeling	
Witbooi (2021)	Local population
and migrant sub- population	SEIR (Susceptible, Exposed,
Infectious and Recovered)	Demonstrated way to quantify the rate of removal of
migrants out of the population after a short visit	
Zheng et al. (2022)	Refugee camps	SEIR SEAIRD	Double-dose vaccination strategy can reduce infection and death, while the single-dose vaccination strategy can postpone the infection peak more efficiently.	

Among the 19 operational studies, specific strategies were outlined for defining and improving vaccination approaches for refugees, as presented in Table 3. On the other hand, the idea of fairness varied greatly among studies. While some discussed fairness regarding community involvement and the use of health services, others emphasized the necessity for a just distribution of vaccines. While in some papers, fairness was described in terms of vaccination hesitancy and vaccination, in one study, fairness was explored as a method to protect vulnerable groups concerning the rate of transmission of infection. The case studies underlined the importance of providing refugees with vaccines but made no specific recommendations on implementation (Table 3).

Several studies highlighted community-based and culturally tailored approaches as critical facilitators. Streuli et al. (2021) integrated community engagement at every stage of an innovative vaccine education technology for Somali refugees, ensuring that materials were culturally and linguistically appropriate to address low health literacy and medical mistrust. Similarly, Kowal, Jardine & Bubela (2015) working with community partners, used informal, language-appropriate communication methods and materials, while Allen et al. (2022b) reported an 87% vaccination completion rate in a study involving collaboration with community leaders and social media groups.

Modeling studies provided valuable insights into fairness-aware strategies. One study used a refugee camp setting to develop single-dose and two-dose outbreak control scenarios. The SEAIRD model, which explored optimal vaccination strategies under the practical constraints of limited medical resources, found that outbreak control in challenging environments such as refugee camps also depends on maximum daily vaccination capacity. Another example is Witbooi’s (2021) SEIR model, which integrated local and migrant populations into a single framework to assess migrant dynamics in terms of inflow and outflow. This model also emphasized the necessity of achieving 95% measles vaccination coverage among newborns in each group.

Barriers and facilitators

In our review, we identified several common barriers and facilitators affecting vaccine uptake among migrants. One of the most prominent barriers was uncertainty about legal status, particularly for undocumented migrants, who either could not access healthcare services or avoided them due to fear of potential consequences. Language barriers were another major challenge, limiting access to the healthcare system and making it difficult to understand vaccination-related information. A general lack of trust in government institutions and the healthcare system further discouraged participation in vaccination programs. Insufficient or misleading information about vaccines also contributed to hesitancy. Logistical difficulties, such as long distances to vaccination centers and limited transportation options, were especially common in rural and peri-urban areas. In addition, cultural and religious beliefs were sometimes linked to skepticism or hesitancy toward vaccines.

On the other hand, several facilitators were reported to improve vaccine access and acceptance. Community-based approaches, including outreach through local leaders and community partners, played a key role in building awareness and trust. Providing multilingual information materials and incorporating cultural sensitivity into campaign design—ensuring that both messages and delivery methods respected local norms and values—were associated with higher participation rates.

In some studies, digital health tools were particularly effective; for example, Knights et al. (2021) noted that language obstacles and challenges in establishing trust limited the effectiveness of digital health messaging, while Streuli et al. (2021) successfully addressed these issues through bilingual virtual technology. Other studies used digital health platforms to conduct surveys and simplify data collection, whereas Streuli et al. (2021) combined cultural elements with community engagement in virtual education, further expanding the applicability of such tools. Finally, digital health tools, such as targeted health messages and visual educational resources, were effective in delivering accurate vaccine information and reaching otherwise hard-to-access groups.

Discussion

We investigated the fairness-based strategy of vaccination in the articles included in this study to understand the vaccination status of migrant groups during pandemics, vaccination policies on migrants, and vaccine distribution models specific to migrants. At the same time, by questioning how migrants access a fair immunization system, we try to understand better the present time and how to be prepared for possible future pandemics. In this sense, the fairness-based dynamics of vaccination are investigated in the articles included in this study.

Unfortunately, many vaccine distribution articles do not focus on migrants. Thus, after full-text readings, only 19 were included in the study, involving populations such as refugees, migrants, immigrants, and migrant groups living with the host population. The distinctions between indigenous residents and migrants—such as significantly lower vaccination rates and higher infection risks among migrants—imply that fairness, as opposed to equality, is more fitting for migrants. Thus, fairness-based approaches in this study refer to process-level justice that supports equity in outcomes, especially for marginalized and mobile groups (Dai et al., 2025). The goal of equity is to compensate for disadvantages and ensure everyone has the opportunity to be healthy. This often requires allocating more resources to underserved or high-risk communities, even if it results in unequal distribution. Fairness, on the other hand, involves ensuring that vaccination policies are applied consistently, free from bias or favoritism, and are perceived as just by the public. While equitable policies seek to achieve equal health outcomes, fair processes strive to ensure that the criteria and procedures behind vaccine allocation are morally and socially acceptable (Prentice et al., 2024; Persad, Peek & Ezekiel, 2020).

Besides similar vaccination rates and vaccination intention, there was no specific systematic and national vaccine distribution program for migrants (Salibi et al., 2021; Streuli et al., 2021; Kowal, Jardine & Bubela, 2015; Hui et al., 2018). This distribution was on a smaller scale and limited. However, inclusive national strategies aiming to ensure that migrants have equitable access to vaccination and achieve coverage levels comparable to the host population have been implemented. Although the literature review was conducted worldwide, all studies represented high and middle-income countries receiving immigrants (Streuli et al., 2021; Kowal, Jardine & Bubela, 2015; Hui et al., 2018; Zhang et al., 2021; Knights et al., 2021). This focus is unsurprising given that these settings typically have more established research infrastructures and accessible published data (Charania et al., 2020). However, it does not fully capture the complexity of global migration patterns, including the significant and growing flows of migrants and refugees between developing countries. This may be due to the fact that vaccination strategies implemented in low-income countries are not published in the academic literature, and it may also reflect a limitation of our study, which included only articles published in English, potentially preventing access to relevant strategies documented in other languages. However, the findings—although limited to studies from high and middle-income countries—highlighted that current vaccination patterns fall short of suggestions for fairness-based immunization strategies that provide priority to immigrants.

While the vaccination rates reported in the articles varied, the intention to receive a vaccine was reported to be high. This disparity may be attributed to limited access to healthcare services, despite a strong willingness to be vaccinated. Most importantly, higher infectivity and lower vaccination rates have been reported among migrants and refugees compared to the local groups (Alabdulla et al., 2021; Diaz, Dimka & Mamelund, 2022; Al-Hatamleh et al., 2022; Zheng et al., 2022). The reasons for this are listed as poor household conditions, crowded living quarters, inadequate access to water, sanitation and hygiene, difficulty in accessing information, language barriers and the effect of cultural elements (Perez-Brumer et al., 2021; Buonsenso & von Both, 2022; Matlin et al., 2022; Jawad et al., 2021). The problem areas created by these factors, in which poverty is a determinant, also complicate the standardization of access and distribution of the vaccine.

Undocumented migrants were included in certain research, while recent arrivals or migrants on the move were discussed in others. In the studies examined, we attempted to form migrants as a part of the distribution by using various methods, which resulted in successful outcomes such as the inclusion of undocumented migrants, use of digital tools, bilingual virtual technologies and community engagement strategies. However, its applicability for non-resident and undocumented migrants is limited. While community participation initiatives can strengthen the concept of justice, they may also complicate implementation in contexts with high population mobility, necessitating adjustments to vaccination strategies. Similarly, although digital tools and bilingual virtual platforms are designed to promote inclusiveness, limited access to these technologies among disadvantaged groups can inadvertently exacerbate existing inequalities. For example, Knights et al. reported that newly arrived migrants face challenges related to digital literacy, language barriers, trust-building, and the risk of missing safety information during virtual consultations (Knights et al., 2021).

The results show considerable variability, with vaccination coverage ranging from approximately 47.2% to 88%, highlighting disparities within and between these groups. Migrant groups tend to exhibit slightly higher vaccination rates compared to refugees. This result emphasizes the challenges in achieving consistent vaccine uptake among displaced populations, underlining the importance of targeted interventions to improve equitable access to vaccines in these vulnerable communities. As a result, healthcare access was the most crucial conclusion considered in these fairness-based investigations. More research is needed on national policies and vaccine delivery models that assess migrants and refugees on a case-by-case basis, indicating an important attitude of neutrality towards immigration status or exclusion (Hui et al., 2018). However, our findings provide a critical comparative foundation from which the policies influencing migrant health in different contexts may be further assessed.

The SEIR model was applied in two of the study’s modeling studies (Zheng et al., 2022; Perez-Brumer et al., 2021; Buonsenso & von Both, 2022; Matlin et al., 2022; Jawad et al., 2021; Witbooi, 2021). Although both are founded on migrants, they could not establish fair vaccination distribution criteria. Mathematical modeling can shed light on the best allocation plans that optimize the advantages of each dose (Joshi et al., 2023). However, such models should be supported by some criteria that minimize inequalities in access and ensure allocation fairness. The conditions that lead to vulnerability in the context of social determinants of health could not be identified in the SEIR model, despite it being a successful model to calculate the changes in the number of susceptible, exposed, infectious and removed individuals in epidemic control. Specific access requirements are necessary when considering how difficult it is for migrants to receive healthcare. Mobile facilities may be utilized to provide access and to plan effective and fair schedules, and optimization models are required. For example, a study conducted with local communities developed a fairness-based SEIR framework for idea dissemination in multilingual environments, integrating social reinforcement, forgetting mechanisms and cross-contagion, which could be adapted to vaccine distribution strategies for migrant populations (Dong et al., 2024). Similarly, in Argentina, an SEIR-based age-group vaccination strategy highlighted the importance of identifying vulnerable groups when vaccine supply is scarce, demonstrating that vaccinating those over 60 years of age significantly reduced mortality (Inthamoussou et al., 2022).

Strengths and limitations

Our study was undertaken during the COVID-19 pandemic, which has raised questions about how quickly vaccines should be distributed. One of the study’s limitations is the inability to assess long-term trends in vaccine dissemination, as most included studies focused on short-term or early-stage interventions. The results of each study alone helped understand vaccination. Still, it was difficult to draw firm conclusions and synthesize results due to the variety of methodologies and methods used to measure rates of vaccination, vaccine acceptance and vaccine requests. Our study faced challenges in identifying publications that specifically addressed fairness vaccine allocation or centered discussions on large-scale migrant vaccination initiatives, as vaccine distribution models have not received significant public attention. Another limitation of this study is its reliance on smaller-scale interventions, which constrained its ability to provide a comprehensive definition of fairness in vaccination. Furthermore, significant heterogeneity across studies—in terms of methodology, outcome measures (e.g., vaccine uptake, acceptance or demand), and target populations—posed challenges for synthesis and comparability.

Another notable limitation is the potential publication bias, as most included studies originated from high and middle-income countries. Government-led or community-based strategies in low-income countries may exist but are unpublished in academic sources. Additionally, the restriction of English-language publications may have further limited the geographic and linguistic diversity of the review.

The review also faced challenges in identifying studies that explicitly addressed fairness-based vaccine allocation or large-scale national immunization efforts targeting migrant populations. Most of the studies focused on smaller-scale interventions or community-level strategies, which constrained the analysis and limited the generalizability of findings.

Despite these limitations, the review contributes to the literature by highlighting the need for consistent terminology when studying displaced populations, and by identifying the gap in research focusing specifically on fairness-oriented vaccination strategies for migrants.

Conclusions

In addition to being an issue of social justice and a rights-based approach to health, fully integrating migrants and refugees into public health efforts is also a practical public health policy required to avoid pandemics. Studies have revealed that, despite using diverse approaches, immigrants and refugees have a high intention to vaccinate, with a lack of awareness and an inadequate comprehension of public health vaccination strategies. Modeling studies identified the best method based on the population and the intensity of the epidemic. Still, a notion of justice that would account for the disadvantages of migrants and refugees has not been developed. Studies evaluating the fairness of vaccine allocation in migrant populations during pandemic periods are scarce. Although studies on access and acceptance of vaccines are intense, more data is needed to understand their usefulness in ensuring fair access.

Supplemental Information

Supplemental Information 1 PRISMA checklist

We would like to express our sincere gratitude to TUBITAK for their invaluable support during the data collection process and their assistance in overcoming technical challenges throughout the preparation of this project.

Additional Information and Declarations

Competing Interests

Author Contributions

Data Availability

The authors declare there are no competing interests.

Sera Şimşek conceived and designed the experiments, performed the experiments, analyzed the data, prepared figures and/or tables, and approved the final draft.

Sevval Altay conceived and designed the experiments, performed the experiments, authored or reviewed drafts of the article, and approved the final draft.

F. Sibel Salman conceived and designed the experiments, authored or reviewed drafts of the article, and approved the final draft.

İlker Kayı conceived and designed the experiments, performed the experiments, analyzed the data, authored or reviewed drafts of the article, and approved the final draft.

The following information was supplied regarding data availability:

This is a literature review.

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
