# Peer review of "Fairness-based techniques to optimize vaccine allocation among migrants during pandemics: a scoping review"

_PeerJ, doi:10.7717/peerj.20208_

## Round 0.1 · original submission · Major Revisions

Thank you for your submission. The reviewers commented positively on the relevance and importance of your review, but had concerns about your terminology, details of your methods, and the breadth and depth of your results and discussion.

Key points for you to address in your revision include:

1. Review the terminology that you use to describe your population of interest, which is in many respects broader than "refugees". (R2)

2. Expand your definition of "vaccine fairness", which would benefit from further description, and its relationship to "equity". (R1 & R2)

3. Clarify details of your methods. (R2)

4. Broaden and deepen your results and discussion. (R1 & R2) The reviewers have provided several excellent suggestions for how your manuscript could be enhanced with additional results and interpretation. I refer you to their detailed comments for specific suggestions.

5. Distinguishing between primary and secondary sources (such as reviews) is critical. You have the choice of either excluding the latter (as suggested by R1) in order to focus on the operational studies, or of retaining both classes, but ensuring the result clearly distinguish between these two categories (as suggested by R2).

6. Ensure that your results and discussion clearly address and match the stated aim of your review.

7. Review your manuscript for grammar and consistency of capitalisation and acronyms, ensuring that all acronyms are defined at point of first usage.

**Language Note:** The review process has identified that the English language must be improved. PeerJ can provide language editing services - please contact us at [email protected] for pricing (be sure to provide your manuscript number and title). Alternatively, you should make your own arrangements to improve the language quality and provide details in your response letter. – PeerJ Staff

Reviewer 1 ·

Basic reporting

This paper would benefit from a close grammar read by a colleague who is fluent in English. There are many grammar inconsistencies and awkward English phrasings throughout the paper that detract from the flow of the paper and the scientific findings. Examples would be line 23 (2000s) or line 128 (“taken as a basis”). Additionally, double-check that acronyms and capitalization are accurate and consistent throughout the article. An example would be lines 98 and 99 (SIR vs SEIR).

The paper discusses the key approaches to vaccine fairness, but a few sentences to deepen the reader's understanding of vaccine fairness would help the reader have clarity early in the manuscript and expand on the need for this review. I would suggest adding more information at the end of the paragraph that ends on line 66. Otherwise, the introduction sets up the purpose of the review study.

Experimental design

The authors have a well-developed aim stated in the introduction. The paper is structured in a logical and coherent way. The headings/subheadings make sense. Authors followed standard review procedures (ECLIPSE, PRISMA, and Checklist for Qualitative Research). The included PRISMA checklist was complete and useful in assessing the validity/rigor of the study. The use of two authors to synthesize the data and discussions of joint review was present in the methods section, adding to the validity of the findings. The eligibility criteria are listed and clear. Only operational articles (n=19) were included in a table (Table 3), but summary tables (Tables 1 & 2) were helpful in understanding the other articles included within the review.

However, this paper would benefit from narrowing the inclusion criteria to only include the 19 operational studies. It is not uncommon within review studies to exclude other reviews, policy papers, grey literature, etc. This change would strengthen the paper and allow the authors to focus more on synthesizing just the operational findings.

Validity of the findings

The benefit to literature is clearly stated. Discussion and conclusion link to the original research question. The authors note the small amount of literature as a limitation. Although the number of articles that discussed fairness allocation modeling was very limited in this population, the discussion could benefit from mentioning how vaccine fairness allocation modeling is done in other populations to give the reader a comparison group and make the discussion section more robust.

It’s clear from the findings of the review that gaps remain in fairness-based techniques for allocating vaccines to refugees. More explicitly stated unresolved questions and how future research could address these unresolved issues as part of the discussion and conclusion.

Additional comments

While there have been plenty of review papers examining vaccination in refugees, this paper offers a unique perspective focused on fairness techniques when allocating vaccines to refugees. The authors did a good job explaining their methodology, which is consistent with the rigor expected of review papers.

I recommend removing the 28 reviews/reports from the body of the assessed literature. This would allow the authors to expand much more on the findings of the 19 operational studies. This could allow a deeper discussion section with more expanded information about the 19 studies and fairness in allocating vaccines to this population.

A review paper with a very limited number of articles can be challenging, but I think the lack of published scientific literature speaks to the need to explore this gap in vaccine distribution. The authors mentioned this difficulty in their limitations section. However, this paper could benefit from expanding the discussion section to include how fairness allocation occurs in other at-risk populations as a comparison point to what has been done elsewhere.

I would also recommend expanding the introduction with several additional sentences explaining vaccine fairness allocation (at the end of paragraph at line 66). This addition would provide more context for the reader moving into the discussion section.

Lastly, the paper would benefit from a close reading by a person who speaks/writes English fluently. There are a number of awkward English phrasings and small grammar mistakes throughout that take away from the reader’s experience.

·

Basic reporting

Thank you for the opportunity to review the manuscript titled "Fairness-based techniques to optimize vaccine allocation among refugees during pandemics: a scoping review." Below, I provide my comments following the structure suggested by the journal.

The article is highly relevant, not only because of its focus on a particularly vulnerable population, but also due to its critical analysis of how rhetoric around equity and justice is often incorporated without being translated into concrete measures within plans, programs, and interventions, such as vaccine allocation for refugees during crises like a pandemic. This scoping review offers an important reflection on the rhetorical use of principles that should, in fact, guide decision-making processes in public health.
Given its relevance and potential contribution to the field, I provide below a series of comments aimed at strengthening the clarity, conceptual consistency, and methodological alignment of the manuscript.

Abstract
The research question stated in the abstract: "What is the impact of fairness-based techniques in optimization of vaccine delivery during pandemics in urban or rural areas on the vaccination rate among refugees?" (lines 15 and 16) frames the review in terms of assessing impact. However, the article primarily focuses on describing the strategies used and reporting outcomes such as vaccination coverage, rather than evaluating broader or final impacts typically associated with impact assessments (e.g., reductions in morbidity or mortality). I suggest revising the research question to better reflect the actual scope of the review, which is more aligned with analyzing approaches and their immediate results rather than their broader health impact.

Introduction
The definition of "refugee" provided in the manuscript: "an 'imperative' migration due to economic and political conditions..." (lines 47-49) does not align with the internationally recognized legal definition established in the 1951 Refugee Convention. That definition, as upheld by UNHCR, refers to individuals who "owing to well-founded fear of being persecuted for reasons of race, religion, nationality, membership of a particular social group or political opinion, is outside the country of [their] nationality and is unable or, owing to such fear, is unwilling to avail [themself] of the protection of that country; or who, not having a nationality and being outside the country of [their] former habitual residence, is unable or, owing to such fear, is unwilling to return to it." (see https://www.unhcr.org/about-unhcr/overview/1951-refugee-convention). The citation used (UNHCR’s “Figures at a Glance” page) does not provide a conceptual definition and therefore does not support the statement. Additionally, the description given in the article more closely resembles the concept of forced displacement, which encompasses broader drivers of migration. I recommend that the authors revise this section to reflect the legally accurate definition of refugee and clarify the distinction between refugees, asylum seekers, and other categories of forcibly displaced persons.

This conceptual inaccuracy may have broader implications throughout the manuscript, as it could influence how the target population is framed, how evidence is interpreted, and how conclusions are drawn. I recommend reviewing the use of the term “refugee” consistently across the text to ensure conceptual and legal accuracy.

The sentence in lines 63–65 is somewhat problematic, as it suggests that individual characteristics of refugees (such as place of birth, cultural beliefs, or language proficiency) are the reason service delivery is difficult. However, it may actually be the opposite: the presence of systemic, structural, and contextual barriers to accessing health services for refugee populations with these characteristics may result in lower immunization coverage. In other words, these attributes may not inherently hinder service delivery, but rather reflect the populations most affected by inequitable or inadequate service provision. I recommend rephrasing this sentence to avoid potential misinterpretation or unintended stigmatization.

Experimental design

The first part, related to context (Lines 82–113), would be more appropriately placed in the Introduction rather than in the Methods section.

Regarding the terminology (Lines 85–86), the authors attempt to distinguish the term fairness from equity, yet this distinction is not well justified. For many authors, equity in health can be defined as fairness in health. The authors should explicitly explain why they consider these concepts separate and provide an appropriate bibliographic reference to support this perspective.

In the Methods section (Lines 166–168), the authors state that records were excluded by “automation tools,” but the process is not clearly described. I recommend specifying which tool or software was used, how it was applied, and what parameters or criteria guided the automated exclusion. This level of detail is essential to ensure transparency and reproducibility.

In the title and abstract screening phase (Lines 168–169), it is unclear how many reviewers were involved or whether the screening was performed independently by multiple reviewers to reduce bias. The authors should specify the number of reviewers and clarify whether the process was done independently, with subsequent consensus, or collaboratively.

In Lines 170–172, the statement “47 articles, the common intersection set of 60 articles in total” is confusing. Does this mean that 60 articles passed screening, but only 47 met the inclusion criteria? Additionally, it is not specified how many reviewers participated in this step. Both the numbers and the process should be clarified.

In Lines 175–182, the classification of the included studies (e.g., 19 operational studies vs. 28 reviews/viewpoints/reports) appears to belong in the Results section rather than the Methods. The Methods section should focus on describing how studies were identified, screened, and classified, whereas reporting the counts or categories of included studies is part of the findings.

Validity of the findings

The Results section does not fully address the stated aim of the review (lines 75–79), which was to identify fairness-based vaccination strategies, vaccination frequency among refugees, vaccination policies during pandemics, and vaccine distribution models specific to refugee populations.
Currently, there is no clear narrative or analytical thread guiding the reader through the findings. The section remains largely descriptive, focusing on term frequency and methodological aspects (e.g., study design), rather than presenting a meaningful synthesis of strategies or fairness-based approaches.

The results also fail to differentiate between findings from original studies and those from reviews. This distinction is critical to understanding how primary evidence versus synthesized knowledge contributes to the analysis of vaccination strategies and outcomes. Presenting the findings separately for these two groups would improve clarity and enhance the reader’s understanding of the evidence base.
The review does not sufficiently analyze or discuss how fairness is conceptualized across the included studies, despite this being central to the research question. A thematic synthesis or conceptual mapping of how fairness is understood and operationalized is necessary to meet the review’s stated objectives.

Furthermore, there is no bibliometric analysis (e.g., timing and geographical context of the studies), and the countries or regions of origin of the migrants and refugees studied are not reported. The current emphasis on word frequency (e.g., “migrant,” “fairness,” “acceptance”) is insufficient to convey the strategies, barriers, and policies that the review aimed to explore.
To strengthen this section, a reorganization and expansion are recommended. Incorporating descriptive subheadings (e.g., Characteristics of included studies, Fairness-based strategies, Barriers and facilitators, Vaccination outcomes) would create a logical structure. Additionally, the inclusion of visual tools (such as summary tables, conceptual figures, or thematic maps) would significantly improve readability and comprehension.

Discusion
In the Discussion (lines 235–243), there is a lack of logical connection between the first part of the paragraph (which explains the inclusion of migrants and immigrants due to terminology issues) and the second part, which moves abruptly to statements about fairness-based approaches and terminology.

Additionally, there are conceptual concerns:
• The distinction between migrants and refugees is oversimplified.
• The assertion that fairness is inherently more suitable than equality when discussing vaccine inclusion for refugees is not well substantiated and does not reflect the complexity of these concepts. In some situations, equality can actually represent the fairest and most equitable approach (e.g., during the COVID-19 pandemic, where prioritizing all older adults regardless of group affiliation was both fair and equitable given their higher mortality risk).
• The interpretation of equity, justice in health, and equality appears superficial, as though minority groups (e.g., Indigenous peoples, Afro-descendants, migrants, Roma, etc) should always receive differential access to services. Justice, however, is context-dependent and involves treating equals equally and unequals unequally, but only where relevant to the specific risk or vulnerability.

I recommend that the authors revisit this paragraph to strengthen the conceptual discussion and ensure a more nuanced understanding of fairness, equity, and equality.

In lines 245–246, the authors note the absence of specific systematic or national vaccine distribution programs for refugees, but this should not necessarily be viewed as a negative finding. It is unrealistic to expect separate vaccination programs for every subgroup of interest, as this would not be feasible for any public health administration. What matters is the existence of inclusive strategies ensuring that migrants and refugees have equitable access to vaccination and achieve coverage levels comparable to the rest of the population. The text could acknowledge this point to provide a more balanced interpretation of the results.

In lines 247–249, the statement oversimplifies migration dynamics. Migration or refugee movements are often not a matter of free choice but are shaped by the opportunities, constraints, and risks people face, including forced displacement. Moreover, the sentence seems to overlook the growing importance of South-South migration flows, which have increased significantly in recent decades. I recommend rephrasing this part of the discussion to reflect a more accurate understanding of global migration dynamics and to avoid oversimplifications.

In lines 249–252, the fact that all included studies come from high- and middle-income countries likely reflects a publication bias rather than indicating that fairness-based vaccination strategies are absent in low-income countries. Many government-led strategies in these settings are not published in academic journals. Moreover, the review was limited to studies published in English, which further restricts the scope and diversity of the findings. I recommend that the authors acknowledge these limitations to avoid overgeneralization and to reflect a more realistic global perspective.

In lines 264–276, the content highlights specific findings, such as the inclusion of undocumented immigrants, use of digital tools, bilingual virtual technologies, and community engagement strategies, that should have been presented in the Results section. Their absence there is a sign that the Results section is underdeveloped and lacks the level of detail expected for a scoping review. I recommend expanding the Results to incorporate this type of information in a structured way (e.g., key strategies, tools, and interventions identified), while using the Discussion to interpret and analyze these findings.

In section 4.1 (lines 297–319), the discussion of strengths and limitations is not sufficiently focused on the actual study. While the authors mention a few limitations (e.g., heterogeneity of methodologies and difficulty drawing firm conclusions), much of the text shifts toward a theoretical discussion of fairness, horizontal vs. vertical equity, and related economic measures (e.g., the Gini index). This theoretical background could be valuable in the introduction or discussion, but here it distracts from a clear, critical assessment of the study’s strengths and weaknesses. I recommend refocusing this section on aspects such as limitations of the review process (e.g., language restriction to English, publication bias, lack of data from low-income countries, among others), strengths of the approach, and methodological challenges specific to this review.

Additional comments

Throughout the manuscript, the terms refugee and migrant are used interchangeably. Moreover, given that the inclusion criteria for the review covered not only refugees but also migrants, immigrants, and asylum seekers (and considering that while all refugees are migrants, not all migrants are refugees), I strongly recommend revising the title and the manuscript as a whole to more accurately reflect the actual scope of the study. Specifically, the review analyzes strategies for including migrant populations in vaccination programs during pandemics, rather than focusing exclusively on refugees.

---

## Round 0.2 · accepted · Accept

Thank you for your detailed attention to the reviewers' comments, I have reviewed the changes and am satisfied that the manuscript is ready for publication. Well done on a nice piece of work on an important topic.

Reviewer 1 ·

Basic reporting

The authors improved the scientific language and terminology within the manuscript. They also have improved consistency in use of acronyms throughout. This has improved the overall quality of the article. The authors also adjusted their definitions of their studied population, making for an overall clearer argument. The introduction is clear and justifies the need for the review. The deeper discussion of vaccine fairness in the introduction improves reader understanding of the purpose of the review and adds overall clarity. The added references are appropriate for the edited manuscript.

Experimental design

The overall study design remains strong. The changes to the number of articles included is a positive change to the manuscript. The change allowed for more emphasis on synthesizing the operational findings. The addition of the headings in the results section also help add clarity to the findings.

Validity of the findings

The changes to the number of articles included is a positive change to the manuscript. The change allowed for more emphasis on synthesizing the operational findings. The addition of the headings in the results section also help add clarity to the findings. The focus on the 19 articles has improved the discussion section and meets the goals set out in the introduction.

Additional comments

The edits addressed my concerns about the manuscript.